# Knowledge Graph Reasoning with Reinforcement Learning Agent guided by Multi-relational Graph Neural Networks

## Abstract

Reinforcement Learning (RL) has emerged as a highly effective technique in various applications, including Knowledge Graph (KG) Completion. KG Completion involves navigating through an incomplete KG from a source entity to a target entity based on a given query relation. However, existing RL-based approaches only focus on training the agent to move along the graph, seldom take into account the multi-relation connectivity inherent in knowledge graphs. In this paper, we propose a novel approach, Reinforcement learning agent Guided by Multi-relation Graph Neural Network(RGMG). Our approach develop a Multi-relation Graph Attention Network (MGAT) which generate high quality KG entity and relation embedding to help agent navigation. Additionally, we develop a Query-aware Action Embedding Enhancement (QAE) module to strength information contained in action embedding. Experiments on various KG reasoning benchmarks demonstrate that RGMG is highly competitive and outperformed current state-of-the-art RL-based methods in different dataset.

## 1 Introduction

Artificial General Intelligence (AGI) has come a long way, and the development of automated knowledge reasoning is a significant milestone. This technology allows computer systems to infer new facts from their existing knowledge, which is critical for achieving AGI. Recent years have seen the construction of comprehensive knowledge graphs (KGs) such as Yago Suchanek et al. (2007) and Freebase Bollacker et al. (2008), containing vast amounts of well-structured facts for entities and their relations. However, due to their large scale, most KGs remain highly incomplete, making KG completion one of the most challenging tasks for automated knowledge reasoning systems.

To address this challenge, we consider to treat KG completion as a reinforcement learning (RL) problem, training an agent to traverse the graph from a source node to a target node and predict the missing relations between entities. A KG is a multi-relational graph, denoted as $(\mathcal{V}, \mathcal{R})$, with $\mathcal{V}$ representing the set of entities (vertices) and $\mathcal{R}$ denoting the set of relations. Given a source node $v_s \in \mathcal{V}$ and a query relation $q \in \mathcal{R}$, the agent aims to find the target node $v_t \in \mathcal{V}$ such that the tuple $(v_s, q, v_t)$ is originally missing from the KG.

Over the years, numerous approaches have been proposed for knowledge graph completion. Tensor factorization methods such as Bordes et al. (2013), Yang et al. (2014), Trouillon et al. (2016), Dettmers et al. (2018a), as well as multi-relational graph neural networks in Shang et al. (2019), Schlichtkrull et al. (2017), Vashishth et al. (2019), have emerged as popular approaches. These methods aim to represent entities and relations as latent embeddings based on existing KG connections. While they are effective at capturing single-hop KG reasoning, they struggle to predict multi-hop connections. To address this limitation, recent studies have focused on training Reinforcement Learning (RL) agents to search for paths over the graph, enabling them to predict multi-hop relations. Early works such as Das et al. (2017) and M-Walk Shen et al. (2018) employed typical RL techniques such as Monte Carlo Policy Gradient (REINFORCE) Williams (1992) and Monte Carlo Tree Search (MCTS) to tackle the task of KG reasoning.

Recently, Deep Reinforcement Learning (DRL) has achieved significant success in various domains, largely due to the use of Deep Neural Networks (DNNs) as excellent function approximators. It is

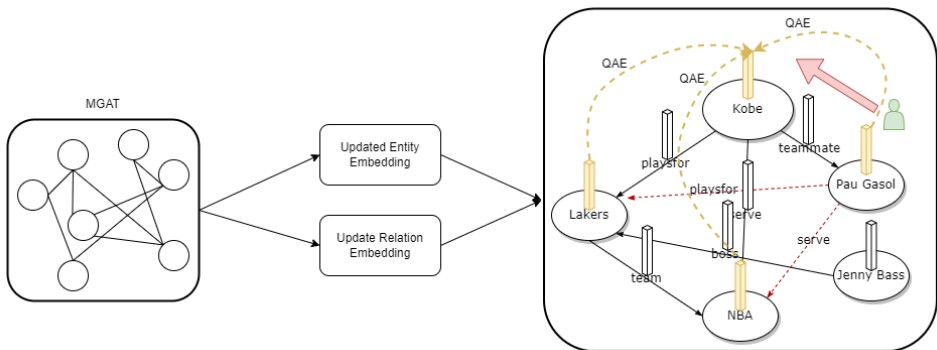

Figure 1: An illustrative overview of our proposed approach, RGMG. During inference time, the MGAT component pre-compute once on all existing KG connections to generate updated entity and relation embeddings. Then for each query, the agent then takes the updated embeddings and performs the QAE component (yellow) to further aggregate the neighbourhood of each possible next action based on the query and state. Finally, the agent (green) decides to move based on the resulting action embedding.

widely believed that when proper RL algorithms are combined with DNNs, computers can achieve promising performance on sequential decision-making tasks, even surpassing human performance, as demonstrated in tasks such as Mnih et al. (2013), Silver et al. (2016). However, most existing RL approaches for KG reasoning fail to integrate KG information with appropriate neural architectures. Instead, they train entity and relation embedding vectors stored in a lookup table as a simple representations, resulting in agents that take actions based on low-quality embeddings trained without using any known graph structural information from the KGs. To enhance the homogeneity of entity and relation embedding, Curl Zhang et al. (2021) proposed grouping pretrained embeddings into clusters and training an additional agent to guide the path between clusters. However, this approach separates the pretraining and clustering of embeddings from the agent learning process, and introducing an extra agent also increases computational costs during inference.

Given the multi-relational nature of KGs, we propose a novel approach for KG reasoning called the Reinforcement learning agent Guided with Multi-relation Graph Neural Network (RGMG). RGMG trains an agent alongside a Multi-relational Graph Attention network (MGAT) in an end-to-end fashion. Through message passing between nodes and edges, MGAT updates entity and relation embeddings based on the existing KG connections. Furthermore, We believe that the way in which neighbouring nodes are aggregated should vary depending on the query relation and agent state. To this end, we developed a Query-aware Action Embedding (QAE) enhancement mechanism, which allows the agent to investigate the neighbourhood of each action node given the query relation and current state. Figure 1 provides an illustrative overview of our proposed approach, RGMG. Finally, our experiments show that the MGAT component generates informative embeddings that are well-clustered and driven by the reinforcement learning objective, while the QAE component accelerates the agent learning process.

## 2 PRELIMINARY

In this section, we first define notations and formulations for the graph reasoning task as a Markov Decision Process (MDP), and then briefly review several previous reinforcement learning approaches to this problem.

### 2.1 DEFINITIONS

Mathematically, recall that a knowledge graph $\mathcal{G} = (\mathcal{V}, \mathcal{R})$ can be described as $\mathcal{G} = \{(v_s, r, v_o), v_s, v_o \in \mathcal{V}, r \in \mathcal{R}\}$, where $(v_s, r, v_t)$ represents a fact tuple. Further denote $X \in \mathbf{R}^{n_v \times d_v}$ and $Z \in \mathbf{R}^{n_r \times d_r}$ be the embedding matrix for entities and relations respectively, then

$x_v \in X$ and $z_r \in Z$ are the embedding vectors of entities $v$ and relation $r$ respectively. The path searching or walking process on KGs can be viewed as a Markov Decision Process (MDP), which is a tuple $(\mathcal{S}, \mathcal{A}, \mathcal{T}, R)$.

For state space $\mathcal{S}$, each state at time step $t$ is defined as $s_t = (\mathcal{G}, (v_s, r_q), v_t) \in \mathcal{S}$, where $\mathcal{G}$ is the knowledge graph with all **known** connections, $v_s \in \mathcal{V}$ is the starting node, $r_q \in \mathcal{R}$ is the query relation and $v_t \in \mathcal{V}$ is the node where agent is standing at time $t$.

For action space $\mathcal{A}$, the collections of possible actions in time step $t$ is defined as $\mathcal{A}_t = \{(v_t, r, v, \mathcal{N}_v) : (v_t, r, v) \in \mathcal{G}, r \in \mathcal{R}, v \in \mathcal{V}\}$, where $\mathcal{N}_v = \{(u, r) : u \in \mathcal{V}, r \in \mathcal{R}, (v, r, u) \in \mathcal{G}\}$ is the neighboring node and relation pair each action in $\mathcal{A}_t$.

For the transition function $\mathcal{T} : (\mathcal{S} \times \mathcal{A}) \rightarrow \mathcal{S}$, describe the dynamics of the interactions between the agent and the environment, transit from one state to next state by taking actions. That is, the agent move from one node at time step $t$ to another node at time step $t + 1$ by taking the relation edge between two node as action.

For the reward function $R$, the agent will receive terminal reward 1 if it reach the target node at termination step $T$ otherwise it will receive zero reward.

## 2.2 RELATED WORKS

Multi-relational Graph Convolution Networks (MGCNs) have been developed in various versions to model graphs with multiple types of connections, such as Knowledge Graphs (KGs). These models are typically based on the message passing framework, which involves aggregating neighbourhood information of each node to generate more homogeneous entity and relation embeddings for downstream tasks. CompGCN Vashishth et al. (2019) proposed a general form of aggregating message passing as follows:

$$x'_v = \sum_{(u,r) \in \mathcal{N}_v} W_v \phi(x_u, z_r)$$
$$z'_r = W_r z_r \tag{1}$$

where $x'_v \in \mathbf{R}_v^d$ and $z'_r \in \mathbf{R}_r^d$ are the updated hidden component of entities and relation embedding for the next graph convolution layer. After several layers of convolutions, the resulting aggregated entity and relation embedding will pass to Multi-Layer Perceptrons (MLPs) to perform predictions for different tasks, such as link prediction and node classification.

Completing missing connections in KGs can also be achieved through logic reasoning using a reinforcement learning agent. Various methods have been proposed to train such an agent, which can traverse the graph and find the target node given a starting node and a query relation. One such method is MINERVA Das et al. (2017), which utilizes a policy network consisting of Long-Short-Term Memory (LSTM) Hochreiter & Schmidhuber (1997) to update the agent's state. It then calculates the probability distribution for the next actions by performing a dot product operation between the state and action embeddings. Mathematically, this approach can be expressed as follows:

$$h_t = \text{LSTM}(h_{t-1}, [z_{r_{t-1}}; x_{v_{t-1}}]$$
$$d_t = \text{softmax}(\mathbf{A_t}(W_2 \text{RELU}(W_1[h_t; z_q; x_{v_{t-1}}]))) \tag{2}$$
$$A_t \sim \text{Categorial}(d_t)$$

where $h_t$ is the updated hidden agent state at the time step $t$, $\mathbf{A_t} \in \mathbf{R}^{|\mathcal{A}_t| \times (d_r + d_v)}$ is defined to be the stack of the embedding of all possible action $a_1, a_2, ..., a_{|\mathcal{A}_t|}$. Each $a_l$ is the embedding vector of an action in the set of possible actions, defined as $a_l = [z_{r_l}; x_{v_l}]$ where $[;]$ is the vector concatenation operator, $v_l$ is the next possible node and $r_l$ is the relation of the edge connecting $v_{t-1}$ and $v_l$. To obtain the probability distribution for next action, a dot product will be operated between $\mathbf{A_t}$ and a encoded vector from agent hidden state $h_t$, embedding of the query relation $z_q$ and $x_{v_{t-1}}$, the entity embedding of the node where the agent stayed at time step $t - 1$. Moreover, the RL agent in MINERVA is trained using the REINFORCE algorithm Williams (1992) with sampled graph walking paths. After MINERVA, several variants have been proposed, including CURL Zhang et al. (2021). They clusters the nodes in KGs based on pretrained embeddings and trains a pair of agents, namely GIANT and DWARF, to guide graph walking between and within clusters.

## 3 PROPOSED APPROACH

The policy networks used in the above-mentioned Reinforcement Learning (RL) approaches mainly consist of LSTMs and MLPs. However, these networks are not well-suited for processing graph data, including Knowledge Graphs (KGs). To fully leverage the information available in KGs and overcome this limitation, we propose integrating Graph Neural Networks (GNNs) and message passing through neighbourhood aggregation into RL agent policy learning. In the following section, we will delve into the details of our proposed approach, building on the notations defined in the previous section.

### 3.1 MULTI-RELATIONAL GRAPH ATTENTION NETWORK

Rather than applying existing Graph Convolution Networks (GCNs), we propose an alternative architecture called the Multi-relational Graph Attention network (MGAT) to encode Knowledge Graphs (KGs) and generate hidden representations for entities and relations based on known information in the KG. In the popular TransE algorithm Bordes et al. (2013), relations are modeled as translation operations on the embedding space between connected entities, with the objective of minimizing $D(x_v + z_r, x_u)$ for all tuples $(v, r, u) \in \mathcal{G}$, where $D$ is a pre-defined distance function. Inspired by the spirit of TransE, MGAT computes attention weights for each neighbouring node by taking the dot product of $x_v + z_r$ and $x_u$. Recall we define the entities embedding matrix as $X \in \mathbf{R}^{n_v \times d_v}$ and relation embedding matrix $Z \in \mathbf{R}^{n_r \times d_r}$, the whole KG node update mechanism is described as follows:

$$
\begin{aligned}
Q(v, r) &= \mathrm{MLP}_q(x_v + z_r) \\
K(u) &= \mathrm{MLP}_k(x_u) \\
V(u, r) &= \mathrm{MLP}_v([x_u; z_r]) \\
att_u &= \mathrm{softmax}(\sum_{(u,r) \in \mathcal{N}_v} \frac{Q(v, r)^T K(u)}{\sqrt{d}}) \\
x'_v &= \sigma(x_v + \sum_{(u,r) \in \mathcal{N}_v} att_u V(u, r))
\end{aligned}
\tag{3}
$$

where $x'_v$ is the updated entities representation for downstream agent decision, $Q, K, V$ are Multi-Layer Preceptrons (MLP) encoder for the query, key and value in attention mechanism, $\sigma$ is any non linearity activation.

Unlike Vashishth et al. (2019) in equation 1, which updates the relation embedding with a linear layer, we also develop a relation update mechanism to generate relation representation as follows,

$$
z'_r = \sigma(z_r + \frac{1}{|M_r|} \sum_{v \in M_r} \mathrm{MLP}_z(x_v))
\tag{4}
$$

where $M_r$ is the set of entities which has at least one connection with other entities in relation $r$.

By utilizing MGAT, our agent is able to obtain global summaries of the existing knowledge graph information in the form of updated entity and relation KG embeddings, denoted as $x'_v$ and $z'_r$, respectively. These updated embeddings are then passed to the latter part of the policy network for further processing. It is worth noting that MGAT is connected to the latter part of the policy network throughout the entire neural network forward pass, and that the extraction of KG information of MGAT is trained according to the REINFORCE loss during the backward pass.

### 3.2 QUERY-AWARE ACTION EMBEDDING ENHANCEMENT

MGAT summarizes the existing connections of the KG and updates the entity and relation embedding. However, we believe that to make informative decisions at each time step, the agent needs to consider the set of possible actions in a query-context-dependent manner, taking into account the query relation and the current state. To address this, we introduce a Query-aware Action Embedding Enhancement (QAE) mechanism. Specifically, we enable the agent to dynamically construct an action neighbourhood aggregation function that is tailored to the query relation and the current agent state. This allows for more nuanced and adaptive decision-making, as the agent can consider

the context of the current query and state. To achieve this, we explore the one-hop neighbourhood of each possible action $a_l$ during the decision-making process. With a bit abuse of notation, we use $a_l$ to represent both action and action embedding,

$$
\begin{aligned}
Q_A(q, h_t) &= \text{MLP}_{q_A}([z'_q; h_t]) \\
K_A(u, r, h_t) &= \text{MLP}_{k_A}([x'_u; z'_r]) \\
att_u &= \text{softmax}(\sum_{(u,v) \in \mathcal{N}_{a_l}} \frac{Q_A^T K_A}{\sqrt{d}}) \\
e_{a_l} &= \sum_{(u,r) \in \mathcal{N}_{a_l}} att_u K_A
\end{aligned}
\tag{5}
$$

where $z'_q, x'_u$, and $z'_r$ are the update embedding obtained from the MGAT component for query relation, neighbouring node $u$ and edge relation $r$ respectively. The resulting $e_a$ is the action enhancement vector. It will be added to the construction of action embedding of $a_l$ in the policy network of RGMG.

### 3.3 POLICY NETWORK

Similar to the MINERVA approach Das et al. (2017), we utilize an LSTM encoder to iteratively encode and update the agent state using equation 6. Additionally, our agent uses the updated entity and relation embeddings $z'rt - 1$ and $x'vt - 1$ generated from the MGAT component. These embeddings are extracted from the underlying knowledge graph and incorporated into the agent state, providing the agent with more comprehensive information for decision-making.

$$
h_t = \text{LSTM}(h_{t-1}, [z'_{r_{t-1}}; x'_{v_{t-1}}])
\tag{6}
$$

After updating the current agent state, the agent will further explore the neighbourhood of each possible action node and obtain an action enhancement vector according via the QAE component using equation 5, and construct the enhanced action embedding as follows,

$$
a_l = \text{MLP}_a([z'_{r_l}; x'_{v_l}; e_{a_l}])
$$

Same as in equation 2, we stack the embedding of the possible actions together, and compute the probability distribution for next action,

$$
\begin{aligned}
d_t &= \text{softmax}(\mathbf{A_t} \text{MLP}_h([h_t; z'_q; x'_{v_{t-1}}])) \\
A_t &\sim \text{Categorial}(d_t)
\end{aligned}
\tag{7}
$$

where $\mathbf{A_t} \in \mathbf{R}^{|\mathcal{A}_t| \times d}$ is the stack of possible action embedding at time step $t$, $\text{MLP}_a : \mathbf{R}^{d^3} \to \mathbf{R}^d$ and $\text{MLP}_h : \mathbf{R}^{d^2} \to \mathbf{R}^d$ are the MLPs to align the hidden vectors from query state vector and actions embedding for dot product operations.

### 3.4 TRAINING

We adopted the Monte Carlo Policy Gradient (REINFORCE) Williams (1992) method to maximize the expected reward for sampled path walking episode, defined as

$$
L(\theta) = \sum_{t=1}^{T} E_\pi[R(S_t, A_t)|S_0]
\tag{8}
$$

To reduce the variance of policy updates and stabilize training, we utilized the moving average of the cumulative discounted reward as the control variate baseline. Additionally, to encourage path diversity, we added an entropy term Haarnoja et al. (2017) to the RL objective during regularization. The entropy term promotes exploration and prevents the policy from collapsing to a single action.

During the training stage, it is crucial to remove the training tuples in each batch from the knowledge graph before passing it to MGAT. This is done to avoid information leakage and ensure that the learned policy is not biased.

During the inference stage, we first perform MGAT in line 5 once to obtain the updated entity and relation embeddings. Then, whenever a query is received, we perform the same operations as described in lines 6-9 and calculate the action probability distribution at each step. Finally, to generate potential trajectories with high probabilities, a beam search is conducted.

Table 1: Description statistics of benchmark datasets

| Datasets | entities | relations | facts (train) | queries (test) |
|---|---|---|---|---|
| WN18RR | 40,945 | 11 | 86,835 | 3,134 |
| NELL-995 | 75,492 | 200 | 154,213 | 39,927 |
| FB15K-237 | 14,505 | 237 | 272,115 | 20,466 |

## 4 EXPERIMENTAL RESULTS

To demonstrate the effectiveness of RGMG, we compared our proposed approach against state-of-the-art KG reasoning baselines on three popular real-world KG datasets: FB15K-237 Toutanova et al. (2015), WN18RR Dettmers et al. (2018b), and NELL-995 Xiong et al. (2017). Our main comparison task in this experiment is query answering in Knowledge Graphs, where the agent is tasked with walking through the knowledge graph to find the most suitable terminal entity $v_T$ such that $(v_s, r_q, v_T)$ is a fact tuple and complete the missing connection, given a source entity and a query relation $(v_s, r_q, ?)$. In the following sections, we discuss the datasets, evaluation methods, implementation, and results in more detail. By doing so, we aim to demonstrate the superior performance of RGMG over existing KG reasoning approaches.

### 4.1 DATESETS

The WN18RR and FB15K-237 datasets were created by removing various sources of test leakage from the original WN18 and FB15K datasets, respectively, thus making the datasets more realistic and challenging. The NELL-995 dataset, released by Xiong et al. (2017), contains separate graphs for each query relation. Following the setup in Das et al. (2017), we combined all the graphs of different relation types into a single graph for the query answering task, considering that some entities may be connected by more than one type of relation. Table 1 presents some descriptive statistics of the knowledge graphs in each dataset. These datasets were selected for evaluation as they are widely used benchmark datasets in the KG reasoning community and have been previously used in several studies.

### 4.2 EVALUATION METHODS

We evaluation our proposed RGMG approach against some state-of-the-art reinforcement learning approaches for Knowledge Graph Reasoning. In particular, MINERVA Das et al. (2017), M-WALK Shen et al. (2018), GaussPath Wan & Du (2021), Curl Zhang et al. (2021). The results for non-RL base approaches are also reported as baseline, including DistMult Yang et al. (2014) and CompIEx Trouillon et al. (2016). For all baseline, we reported and compared with the performance presented in their original paper.

For each test sample in each of the datasets, the trained agent generates a collection of possible walking paths and terminal entity nodes via beam search with a fixed beam width. We report standard ranking metrics for Graph completion tasks, including Hits@1, 3, and 10, as well as Mean Reciprocal Ranking (MRR). These metrics provide an evaluation of the agent's performance in accurately completing knowledge graph queries by ranking the correct terminal entity nodes higher than the incorrect ones.

There are two type of tasks to evaluate the algorithm. Link prediction in KG completion involves predicting missing entities in unknown links of an incomplete KG. In our approach, we utilize multiple rollouts to search for missing entities in queries of the form (e1, r, ?) or (?, r, e2). On the other hand, fact prediction is a related but subtly different task that aims to infer whether an unknown fact (triple) holds or not. According to Xiong et al. (2017), true test triples are ranked with some generated false triples. To evaluate this task, we first remove all links of groundtruth relations in the raw KG. Then, our dual agents attempt to infer and traverse the KG to reach the target entity. We report Mean Average Precision (MAP) scores for various relation tasks of NELL-995 (corresponding to different subsets).

Table 2: Query answering performance measured by Hits@1, 3, 10 and MRR of our approach is compared to state-of-the-art approaches. The best approach in each category is highlighted.

| | WN18RR | | | | NELL-995 | | | | FB15K-237 | | | | |
|---|---|---|---|---|---|---|---|---|---|---|---|---|---|
| Model | @1 | @3 | @10 | MRR | @1 | @3 | @10 | MRR | @1 | @3 | @10 | MRR | Rank |
| DistMult | 41.0 | 44.1 | 47.5 | 43.3 | 61.0 | 73.3 | 79.5 | 68.0 | 27.5 | 41.7 | 56.8 | 37.0 | 4.42 |
| ComplEx | 38.2 | 43.3 | 48.0 | 41.5 | 61.2 | 76.1 | 82.1 | 68.4 | **30.3** | **43.4** | **57.2** | **39.4** | 3.75 |
| MINERVA | 41.3 | 45.6 | 51.3 | 44.8 | 66.3 | 77.3 | 83.1 | 72.5 | 21.7 | 32.9 | 45.6 | 29.3 | 3.75 |
| M-WALK | 41.5 | 44.7 | **54.3** | 43.7 | 63.2 | 75.7 | 81.9 | 70.7 | 16.8 | 24.5 | 40.3 | 23.4 | 4.50 |
| Curl | 42.9 | 47.1 | 52.3 | 46.0 | 66.7 | 78.6 | 84.3 | 73.8 | 22.4 | 34.1 | 47.0 | 30.6 | 2.67 |
| RGMG | **44.1** | **48.7** | 53.3 | **47.2** | **69.8** | **79.6** | **85.5** | **75.6** | 25.8 | 35.4 | 44.2 | 32.0 | **1.92** |

Table 3: Fact prediction performance, MAP scores for different tasks in NELL-995 dataset.

| TASK | DeepPath | MINERVA | M-Walk | CURL | RGMG |
|---|---|---|---|---|---|
| AthletePlaysInLeague | 96.0 | 94.0 | 96.1 | 97.1 | **97.4** |
| AthletePlaysForTeam | 75.0 | 80.0 | **84.7** | 82.9 | 84.0 |
| AthleteHomeStadium | 89.0 | 89.8 | 91.9 | **94.3** | 92.0 |
| TeamPlaysSports | 73.8 | 88.0 | 88.4 | 88.7 | **89.3** |
| AthletePlaysSport | 95.7 | 98.0 | 98.3 | **98.4** | 98.3 |
| OrganizationHiredPerson | 74.2 | 85.6 | **88.8** | 87.6 | 87.7 |
| PersonBornInLocation | 75.7 | 78.0 | 81.2 | 82.1 | **84.2** |
| WorksFor | 71.1 | 81.0 | **83.2** | 82.1 | 82.7 |
| OrgHeadquarteredInCity | 79.0 | 94.0 | 94.3 | **94.8** | 94.1 |
| PersonLeadsOrganization | 79.5 | 87.7 | 88.3 | 88.9 | **89.1** |
| Overall | 80.9 | 87.6 | 89.5 | 89.7 | **89.9** |

## 4.3 IMPLEMENTATION DETAILS

**Message Passing Framework**. We utilized the PyTorch-Geometric library Fey & Lenssen (2019) for message passing operations in node neighbourhood aggregation. This library, built on top of PyTorch Paszke et al. (2019), allowed for easy implementation and training of Graph Neural Networks (GNNs). We relied on a useful pytorch-scatter function to compute matrix operations in the Compressed Row Storage (CRS) format.

**Dataset**. Following the approach of MINERVA Das et al. (2017) and Curl Zhang et al. (2021), we further split the existing fact tuples in the KG into non-overlapping train and validation sets. Specifically, we separated 3034, 543, and 17535 fact tuples from the training facts for use in the validation dataset, which was used for all model selection and hyperparameter tuning. In trainin stage, since our approach summarizes all known connections in the graph using GNNs to aid the agent in path searching, we removed all training tuple samples in each batch from the KG before passing it to the MGAT module.

## 4.4 COMPARISONS WITH STATE-OF-THE-ART

Following the approach of Das et al. (2017) and Zhang et al. (2021), we evaluated our proposed approach against other methods using the same test datasets. Table 2 presents the results of various approaches on the three benchmark datasets. Our proposed RGMG achieved the best overall performance, ranking at 1.92 on average among all the metrics. Notably, RGMG outperformed all existing approaches on the WN18RR dataset, with a significant difference in performance. On the Nell-995 dataset, RGMG achieved better performance on most metrics, except for Hits@10, and was comparable to SOTA models in general.

On the FB15K-237 dataset, among all the reinforcement learning-based models, RGMG achieved the best overall results. However, as observed in Table 2, embedding-based approaches like DistMult Yang et al. (2014) and Complex Trouillon et al. (2016) dominated over RL multi-hop logic reasoning-based methods. Further investigation and reference to Bordes et al. (2013) revealed that

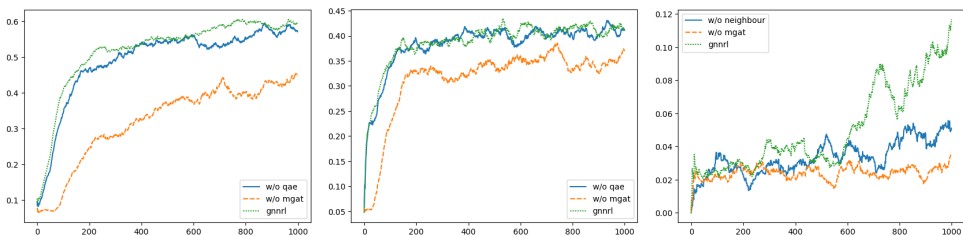

Figure 2: The positive reward rate of different models on the three main test datasets. Left: Nell-955; Middle: WN18RR; Right: FB15K-237

Table 4: Some example paths searched by RGMG

---

**i) About born location**

Politician Ryan $\xrightarrow{\text{PersonBornInLocation}}$ ?(City York)

Politician Ryan $\xrightarrow{\text{PoliticianHoldsOffice}}$ PoliticalOffice Governor $\xrightarrow{\text{PoliticianusHoldsOffice}^{-1}}$ Politicianus Nixon

$\xrightarrow{\text{PersonBornInCity}}$ City York

---

**ii) About sport athlete**

Personus Kobe Bryant $\xrightarrow{\text{AthletePlaysSport}}$ ?(Sport Basketball)

Personus Kobe Bryant $\xrightarrow{\text{AthleteLedSportsTeam}}$ Sportsteam Los Angeles Lakers $\xrightarrow{\text{athleteplaysforteam}^{-1}}$ Athlete Pau Gasol

$\xrightarrow{\text{athleteplayssport}}$ Sport Basketball

---

**iii) About organization hires:**

Sportsteam Florida Gators $\xrightarrow{\text{OrganizationHiredPerson}}$ ?(Visualartist Meyer)

Sportsteam Florida Gators $\xrightarrow{\text{CoachesTeam}^{-1}}$ Coach Urban Meyer $\xrightarrow{\text{OrganizationHiredPerson}^{-1}}$ University Uf

$\xrightarrow{\text{OrganizationHiredPerson}}$ Visualartist Meyer

---

the distribution of query relation types in FB15K-237 is significantly different from that of WN18RR and Nell-995 datasets. Specifically, FB15K-237 contains a larger proportion of 1-to-M relation samples than other datasets. For 1-to-M type query instances, the agent is required to predict a list of entities. However, during the evaluation of multi-hop logic reasoning-based methods, we aim to search for a single target entity. This can lead to the agent getting stuck in some local nodes centered in the target list of entities. Additionally, some target entities of the triple may not have a possible path to travel from the source entity due to the incomplete nature of the knowledge graph.

Our evaluation of fact prediction on the Nell-995 dataset, as shown in Table 3, demonstrates that RGMG performs the best on only 4 out of 10 tasks. However, it achieves the highest overall average MAP scores when compared to other RL-based algorithms. This indicates that RGMG can consistently outperform existing methods, even when evaluating different types of tasks.

### 4.5 FURTHER ANALYSIS

The results presented in the above provide strong evidence that our proposed method outperforms existing approaches. We attribute this success to the efficient learning process of our agent guided by a Multi-relational Graph Attention Network (MGAT) and the Query-aware Action embedding Enhancement (QAE) component that boosts performance further. To validate this hypothesis, we conducted ablation studies on the test dataset to examine the impact of different components of our proposed model.

In Figure 2, we compared the performance of different versions of our model. The model tagged as 'w/o mgat' refers to the same architecture as the proposed model, except that the MGAT part is removed. Similarly, the QAE component is removed for the 'w/o qae' model. Note that only the reward rate in the first 3000 iterations is plotted. From the figures, it is evident that the 'w/o mgat' model requires more iterations to improve the positive reward rate, and its peak reward is

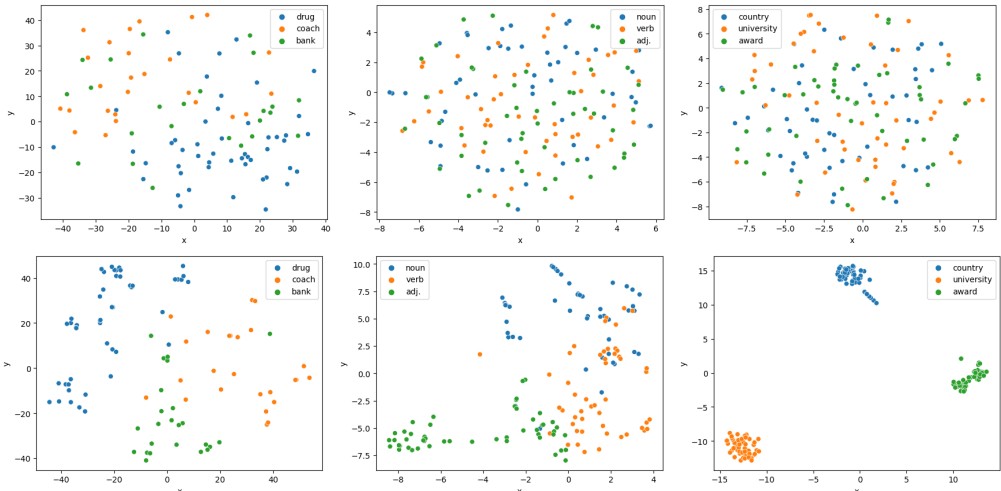

Figure 3: The positive reward rate of different models on the three main test datasets. Left: Nell-955; Middle: WN18RR; Right: FB15K-237

lower than other versions of the model. This demonstrates that adding Graph Neural Networks (GNNs) is beneficial for the convergence of learning process. On the other hand, although the QAE component does not show clear faster convergence, it consistently helps the model reach a higher peak across all the datasets. These quantitative results provide strong evidence for the effectiveness of our proposed method. In addition to the quantitative analysis, we also provide examples of paths discovered by RGMG in Table 4 to demonstrate its ability to find the target node for different types of KG completion tasks.

In addition, we showcase the effectiveness of the Multi-relation Graph Attention Network (MGAT) in generating well-clustered entity embeddings for downstream agent path searching in Figure 3. The top row of the figure shows the TSNE plot of sampled entity embedding before passing through MGAT, while the bottom row shows the embedding after passing through MGAT. For each dataset, we selected entities from three example categories, namely [drug, coach, bank], [noun, verb, adj.], and [country, university, award], for Nell-955, WN18RR, and FB15K-237, respectively. As evident from Figure 3, the embeddings generated from MGAT are grouped with clear boundaries according to their respective categories. These results demonstrate the effectiveness of MGAT in generating well-clustered embeddings, which can significantly improve the agent's performance in searching for the target entities.

Unlike the Curl algorithm proposed by Zhang et al. (2021), which requires an extra GIANT agent to guide the traveling between clusters since the cluster and embedding are predetermined before agent learning, we propose to generate well-clustered embeddings driven by the REINFORCE from knowledge graph structure object via MGAT and train the agent together with MGAT in an end-to-end style. This approach not only produces high-quality and task-oriented embeddings but also avoids the need for extra agent management and resource since all the KG embeddings can be pre-computed once before path search.

## 5 CONCLUSIONS

In conclusion, our proposed approach of training Reinforcement learning agent Guided with Multi-relation Graph Neural Network (RGMG) for Knowledge Graph (KG) Completion has shown promising results. By incorporating a Multi-relation Graph Attention network (MGAT) for KG entity and relation embedding and a Query-aware Action embedding Enhancement (QAE) module for action embedding, our approach has demonstrated competitive performance compared to current state-of-the-art RL-based methods on various KG reasoning benchmarks.

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
