# OpenReview forum: "Knowledge Graph Reasoning with Reinforcement Learning Agent guided by Multi-relational Graph Neural Networks"
_ICLR.cc/2024/Conference — Submitted to ICLR 2024_

### Official Review · Reviewer_SJXD · 2023-10-25

**Soundness:** 2 fair
**Presentation:** 1 poor
**Contribution:** 2 fair
**Rating:** 3
**Confidence:** 4

**Summary:**

This paper introduces RGMG, a novel approach for Knowledge Graph (KG) Completion using Reinforcement Learning (RL). RGMG incorporates a Multi-relation Graph Attention Network (MGAT) to generate high-quality KG embeddings and a Query-aware Action Embedding Enhancement (QAE) module to enhance action embeddings. Experimental results on KG reasoning benchmarks show that RGMG outperforms existing RL-based methods.

**Strengths:**

The performance of the proposed method is promising compared with the RL-based methods.

The idea of query-aware action embedding enhancement is interesting.

**Weaknesses:**

The current draft requires further polishing. The figures need to be generated with higher DPI, and the tables (e.g., Table 1 and Table 3) should be formatted correctly. Overall, the paper is not yet ready for publication.

The novelty of this paper may be limited. As an RL-based method, RGMG only modifies the input embedding modules while keeping the RL module unchanged compared to existing works.

The selected baselines for comparison are outdated, despite being RL-based. In my opinion, RGMG consumes more resources during training and inference. However, its performance is still inferior to state-of-the-art methods that are not RL-based.

**Questions:**

In Figure 2, why is the proposed QAE helpful only on FB15K-237?

---

### Official Review · Reviewer_bkR9 · 2023-10-29

**Soundness:** 3 good
**Presentation:** 1 poor
**Contribution:** 2 fair
**Rating:** 3
**Confidence:** 5

**Summary:**

This paper extends the reinforcement learning based KG reasoning methods with a designed GNN. Instead of using direct embeddings of entities and relations, a multi-relation graph attention network with query-aware action embedding is proposed to learn from the KG structures. With this extension, the proposed method outperforms existing reinforcement learning based methods.

**Strengths:**

1. The idea is easy to capture by incorporating multi-relational GNN to reinforcement learning KG reasoning methods.

2. The proposed method outperforms other reinforcement learning method for KG reasoning.

3. The experiments analyzed several aspects, including general performance, relation-wise performance, ablation study, and path and embedding visualization.

**Weaknesses:**

1. The novelty of the proposed method is weak.
- Using multi-relational GNN to aggregate embeddings in KG is a common choice and direct idea in the literature.
- The main contribution I understand is the query-aware action embedding. However, such an extension is mainly achieved by incorporating the relation embedding into the message-passing functions.

2. The scope of this method is limited.
- Reinforcement learning-based method is out-of-date. The nearest baseline is CURL, which was published in AAAI 2022. The other two methods are MINERVA in ICLR 2017 and M-walk in NeurIPS 2018. The are little related works published in the top conferences (ICML, ICLR, NeurIPS) in recent years.
- The current trend in KG reasoning changes to path-based or GNN-based methods, like NBFNet and A*Net, both of which perform much better than the results in Table 2. In addition, these methods can do inductive reasoning and also generate interpretable paths.

3. The compared methods are not enough. There have been a lot of KG reasoning methods over the last decade, but only five methods (DistMult in 2015, ComplEx in 2017, MINERVA in 2017, M-Walk in 2018, and Curl in 2022) are compared.

4. Many typos:
- The first paragraph in section 3.3, z'rt-1 and x'vt-1?
- Section 3.4, what is line 5? should be Equation 5?
- The values of 3034, 543, 17535 are validation tuples for the three datasets? not clearly written.
- The caption of Figure 3 is identical to Figure 2.
These typos are very obvious and made a very bad impression for me. I think the authors did not take this submission seriously.

**Questions:**

1. Can you compare the proposed method with more recent methods?
2. Can you discuss with the paths-based and GNN-based methods, like NBFNet and A*net, published in the recent years?

---

### Official Review · Reviewer_Sumk · 2023-10-30

**Soundness:** 1 poor
**Presentation:** 2 fair
**Contribution:** 1 poor
**Rating:** 1
**Confidence:** 4

**Summary:**

This paper investigates the KG completion task. A modified GAT attention network MGAT is proposed for multiple relations in the graph. In the experimental parts, the benchmark datasets WN18RR. FB15K-237, NELL-995 are used for evaluation. The performance is comparable with the baseline methods.

**Strengths:**

- The overall structure is good, adding more details would be better.

**Weaknesses:**

- The top concern is the novelty, the framework is an implementation of previous methods.
- The experiments are incomplete. For example, In Section 1 (first paragraph in page 2), it says" However, this approach separates the pretraining and clustering of embeddings from the agent learning process, and
introducing an extra agent also increases computational costs during inference."  However, in the Experiment section, no computational time comparison can be found.

**Questions:**

n/a

---

### Official Review · Reviewer_34kk · 2023-10-31

**Soundness:** 3 good
**Presentation:** 2 fair
**Contribution:** 2 fair
**Rating:** 3
**Confidence:** 3

**Summary:**

This paper propose a reinforcement-learning-based knowledge graph reasoning method guided by multi-relational graph neural networks, called RGMG. RGMG include two novel module, a multi-relation graph attention network (MGAT) and a query-aware action embedding (QAE). Authors evaluate RGMC on three commonly used knowledge graph completion benchmarks and show RGMG achieves relatively good results.

**Strengths:**

1. This work propose to incorporate the graph neural network into the reinforcement-learning-based KG reasoning method, to capture information contained in  the graph structure better, which sounds reasonable.
2. Overall, the proposed method incorporates relational GNN into the reinforcement-learning-based KG reasoning method in a direct way and is easy to understand.

**Weaknesses:**

1. The key motivation of investigating reinforcement learning method for KGC is the embedding-based methods "are effective at capturing single-hop KG reasoning, they struggle to predict multi-hop connections", but there is no experiment related to predict multi-hop connections, thus the motivation is not well supported.
2. The KGE baseline in the experiments are bit old, and recently proposed embedding-based methods such as HAKE, PairRE are not included. And the Section 4.4 is entitled "Comparisons with state-of-the-art" but the state-of-the-art results are not included. And the GNN-based methods, such as CompGCN introduced in the related work should also be regarded as baseline.
3. There are some writing mistakes that should be fixed to improve the overall quality of the paper. Following are some examples:
* In page 5, "The resulting $e_a$ is the action enhancement vector." should be "The resulting $e_{a_l}$ is the action enhancement vector."
* In page 5, The " $z′rt − 1$ and $x′vt − 1$" in "our agent uses the updated entity and relation embeddings $z′rt − 1$ and $x′vt − 1$ generated from" should be fixed.
* In page 5, "line 5" and "lines 6-9" are mentioned in the last paragraph, but I can't figure out what they refer to.
* In page 6, it is mentioned that "out dual agents attempt to infer the traverse the KG to ...", as I understand, there is only one agent in the proposed method, right?
* The caption of Figure 3 is the same as Figure 2.
* Figure 2 shows the 1000 iterations (corresponding to the horizontal axis) while it is said that "only the reward rate in the first 3000 iterations is plotted".

**Questions:**

1. In the introduction, it is said Curl introduced an extra agent and increases computational costs during inference. But the relational GNN introduced into the RGMG also increase the computational costs. So what is the key advantages of RGMG compared to Curl?
2. Does the MGAT module with attention mechanism performs better than original CompGCN?

---

### Meta-Review · Area_Chair_zdjq · 2023-12-07

**Metareview:**

This paper proposes a reinforcement learning agent guided by Multi-relation Graph Neural Network (RGMG) for KG reasoning. The approach develops a Multi-relation Graph Attention Network (MGAT) which generates KG entity and relation embedding to help agent navigation. There are a lot of writing issues, and the reviewers question about the novelty and the compared methods. The paper could be further improved based on the feedback.

**Justification For Why Not Higher Score:**

There are a lot of writing issues, and the reviewers question about the novelty and the compared methods. The paper could be further improved based on the feedback.

**Justification For Why Not Lower Score:**

N/A

---

### Decision · Program_Chairs · 2024-01-16

Reject